# Comprehensive Identification and Characterization of HML-9 Group in Chimpanzee Genome

**DOI:** 10.3390/v16060892

**Published:** 2024-05-31

**Authors:** Mingyue Chen, Caiqin Yang, Xiuli Zhai, Chunlei Wang, Mengying Liu, Bohan Zhang, Xing Guo, Yanglan Wang, Hanping Li, Yongjian Liu, Jingwan Han, Xiaolin Wang, Jingyun Li, Lei Jia, Lin Li

**Affiliations:** 1National 111 Center for Cellular Regulation and Molecular Pharmaceutics, Key Laboratory of Fermentation Engineering, Hubei University of Technology, Wuhan 430068, China; chenmy2007525@163.com; 2State Key Laboratory of Pathogen and Biosecurity, Academy of Military Medical Sciences, Beijing 100071, China; y_17311983041@163.com (C.Y.); zhaixiuli2021@126.com (X.Z.); wangxiaobai1868@163.com (C.W.); ccbbpxx@sina.com (M.L.); zbhforjob@163.com (B.Z.); g110080@outlook.com (X.G.); 3180400053@zju.edu.cn (Y.W.); hanpingline@163.com (H.L.); yongjian325@sina.com (Y.L.); hanjingwan@outlook.com (J.H.); woodsxl@163.com (X.W.); lijyjk@163.com (J.L.); 3Department of Microbiology, School of Basic Medicine, Anhui Medical University, Hefei 230032, China; 4School of Life Sciences, Tsinghua University, Beijing 100084, China; 5College of Life Science and Technology, Beijing University of Chemical Technology, Beijing 100029, China

**Keywords:** endogenous retroviruses, HML-9, BLAT, proviral element, solo LTR

## Abstract

Endogenous retroviruses (ERVs) are related to long terminal repeat (LTR) retrotransposons, comprising gene sequences of exogenous retroviruses integrated into the host genome and inherited according to Mendelian law. They are considered to have contributed greatly to the evolution of host genome structure and function. We previously characterized HERV-K HML-9 in the human genome. However, the biological function of this type of element in the genome of the chimpanzee, which is the closest living relative of humans, largely remains elusive. Therefore, the current study aims to characterize HML-9 in the chimpanzee genome and to compare the results with those in the human genome. Firstly, we report the distribution and genetic structural characterization of the 26 proviral elements and 38 solo LTR elements of HML-9 in the chimpanzee genome. The results showed that the distribution of these elements displayed a non-random integration pattern, and only six elements maintained a relatively complete structure. Then, we analyze their phylogeny and reveal that the identified elements all cluster together with HML-9 references and with those identified in the human genome. The HML-9 integration time was estimated based on the 2-LTR approach, and the results showed that HML-9 elements were integrated into the chimpanzee genome between 14 and 36 million years ago and into the human genome between 18 and 49 mya. In addition, conserved motifs, cis-regulatory regions, and enriched PBS sequence features in the chimpanzee genome were predicted based on bioinformatics. The results show that pathways significantly enriched for ERV LTR-regulated genes found in the chimpanzee genome are closely associated with disease development, including neurological and neurodevelopmental psychiatric disorders. In summary, the identification, characterization, and genomics of HML-9 presented here not only contribute to our understanding of the role of ERVs in primate evolution but also to our understanding of their biofunctional significance.

## 1. Introduction

Endogenous retroviruses (ERVs), discovered in the late 1960s and early 1970s via their derivation from exogenous retroviruses (XRVs) that infected germ lines throughout evolution, and which were subsequently integrated into the genomes of germ cells, eventually coevolved with the host and became fixed in future generations, spreading vertically as proviruses in a Mendelian-inherited manner [1,2,3,4,5,6]. The genomic structure of ERVs is identical to that of retroviruses, comprising internal coding sequences *gag*, *pro*, *pol*, and *env* (5–10 kb in length), flanked by a pair of identical long terminal repeats (LTRs) with cis-regulatory elements (CREs) (300–1200 nucleotides). There are many varieties of retroviruses, generally named by adding letters to indicate the species before the “ERV”. ERVs are classified according to phylogeny, being divided into three major classes. Class I includes Gamma-like and Epsilon-like iterations; class II includes Beta-like versions and lentivirus; and class III includes Spuma-like [7,8] variants. The PBS region of class II ERVs is complementary to lysine (K) tRNA molecules; therefore, these ERVs are named ERV-Ks.

ERVs initially integrate as proviral sequences, ranging from intact proviruses to highly fragmented proviral elements. Over time, they may lose the ability to replicate and transpose through destructive mutations, recombination, methylation, histone modifications, and other host defense mechanisms (thus generating intrinsic immunity to retroviral infection) [9,10,11,12,13,14], with the extent of sequence degradation being roughly correlated with the time of germline insertion of the provirus. Approximately 90% of the ERV sequences in the genome are solo LTRs, which are generated through homologous recombination between 5′ LTRs and 3′ LTRs. This process leads to the deletion of all internal sequences, including viral genes [15,16]. LTRs are highly variable sequences within the retroviral genome, and there is little similarity between retroviral LTRs of different genotypes [17,18]. As a result, annotating solo LTRs in genome assemblies often relies on their association with known retroviruses or previously characterized ERVs. However, there have been reports of the query-independent identification of LTRs [9,17,18].

From an evolutionary perspective, chimpanzees and humans originate from the same ancestor, sharing numerous similar sequences, and the evolutionary patterns of protein-coding genes in the two species are highly related [19,20,21,22,23]. ERV elements and other repetitive elements were initially perceived as parasitic elements whose presence reduced host fitness and were therefore derided as “junk DNA” [24,25]. Now, there is growing evidence that ERVs are widely distributed throughout the entire human and chimpanzee genomes, playing vital roles in genome and gene evolution, epigenetics, and gene regulation [26,27,28,29]. Studies have shown that chimpanzee endogenous retroviruses (CERVs) may be more numerous than human endogenous retroviruses (HERVs) [23]. The chimpanzee genome contains at least 42 independent ERV groups, of which 29 belong to class I, 10 belong to class II, and 3 belong to class III. Among the 42 groups, excluding CERV1/PTERV1 and CERV2, all CERVs were found to display lineal homologues in humans [30,31]. Fewer than 1% of the tested CREs showed different activity between humans and chimpanzees, indicating that small changes in gene regulation can also produce significant differences and confirming that cis-regulatory evolution plays a central role in primate diversity [32,33,34,35,36]. ERVs can enrich for certain active histone modifications and transcription factors and act as cis-regulatory elements to regulate host gene expression [37,38]. Some studies have shown that different types of insertions and deletions are associated with specific epigenomic diversity between humans and chimpanzees [14], and about 7% of chimpanzee–human insertion–deletion (INDEL) variants are related to ERV sequences [39,40]. CERVs are among the factors contributing to INDEL in the human and chimpanzee genomes and account for 7% of the 4% genome-wide difference between the two species [41]. Despite the availability of human and chimpanzee genomes, little is known about the differences in cis-regulation between humans and our closest evolutionary relatives.

The endogenous retrovirus K (ERV-K) is the most recently integrated endogenous retrovirus in the human and chimpanzee genomes, including HML-1–10. HML-9 is an important member of ERV-K family. However, there have been few studies on HML-9. Therefore, developing a precise and updated HML-9 genomic characterization is critical to understanding both the evolutionary history and mechanism effect of these elements in primates. The HML-9 (HERV-K14C) elements entered the reproductive line of primates following the differentiation of the Old World and New World monkey lineages approximately 39 million years ago [41]. Study shows that HERV-K14C-related sequences were amplified during the evolution of the Y chromosome and contributed to the genomic diversity of this chromosome in the great ape lineage [42]. Homologous recombination between HERV-K14C LTRs associated with testis-specific transcripts linked to the Y chromosome (TTY) leads to TTY deletion events, indicating that ERVs may cause genomic instability by inducing new insertions and via the homologous recombination of internal ERV sequences (especially LTRs), leading to deletions. These deletion events may be associated with some cases of genomic instability that result in male infertility [43].

To sum up, the majority of ERV groups almost exclusively comprise solo LTRs, and these genomic structural variants persist in HERVs. Research has documented the presence of HERV-K, older HERV-H, and HERV-W alleles as proviruses in some individuals and as solo LTRs in others [44,45]. In addition, comprehensive analyses of the existence and distribution of HERV-K HML-9 elements in the human genome have been presented in previous studies, along with detailed accounts of the structure and phylogenetic characteristics of this group [46]. However, the biological function of HML-9 elements in the chimpanzee genome remains largely elusive. Hence, we have identified the sequences of HML-9 proviruses and solo LTRs in the chimpanzee genome and conducted a comprehensive analysis of their presence and distribution, describing the structural and phylogenetic properties of the group in detail. Moreover, we have analyzed the integration time of proviruses, genes that may be regulated by these elements, and PBS sequence features. In general, this study intends to provide clear and comprehensive features for HML-9 elements in the chimpanzee genome. The aim is to provide a foundation for more detailed expression studies, the development of which is essential for understanding HML-9′s potential role in physiological and pathological contexts and for conducting further functional studies that focus on specific sites of interest.

## 2. Materials and Methods

### 2.1. Identification, Localization, and Genomic Distribution of ERV-K HML-9 Proviruses and Solo LTR Elements

To determine the chromosomal location distribution of HML-9 proviruses and solo LTRs in chimpanzee genes, we performed HML-9 identification using January 2018 (Clint_PTRv2/panTro6) as a chimpanzee genome reference. Chromosome coordinates were retrieved from the UCSC Genome Browser database (http://genome.ucsc.edu/cgi-bin/hgGateway, accessed on 26 November 2021) using the assembled LTR14C-HERVK14C-LTR14C sequence as a BLAT query in order to collect ERV-K HML-9 sequences from the chimpanzee genome assembly January 2018 (Clint_PTRv2/panTro6). Generally, two resources can be selected as references: consensus representatives or single best-representative strains. The major advantage of using consensus representatives is their much broader representation [47,48]. Therefore, they are utilized as references or queries in most studies. The assembled LTR14C-HERVK14C-LTR14C featured in the current work is drawn from the Dfam database (https://dfam.org/home, accessed on 27 January 2021). The limitations of BLAT search for this analysis mainly include the risk of false positives when a putative hit actually better represents a different ERV. To reduce the risk of false positives from a BLAT search, a phylogenetic analysis would follow to further confirm the assignment of identified HML-9 elements. In addition, the expected distribution of ERV-K HML-9 loci on each chromosome was predicted according to the formula e = Cl × n/Tl (e is the expected number of integrations in the chromosome, Cl represents the unconnected length of the chromosome, n is the total number of actual ERV-K HML-9 loci identified in the chimpanzee genome, and Tl represents the sum of the unconnected lengths of all chromosomes). The obtained number was then compared with the actual number of ERV-K HML-9 sequences, where the former number represented the expected proportion of ERV-K HML-9 insertions per chromosome based on the random distribution principle. The difference between the expected number of integrations and the actual number of integrations was analyzed via the chi-square test (χ^2^), and statistical significance was estimated based on *p*-values.

### 2.2. Structural Analyses

The nucleotide sequences of each identified HML-9 element were compared and analyzed using software and LTR14C-HERVK14C-LTR14C, and all insertions and deletions were annotated in a graphical representation of multiple alignments.

### 2.3. Phylogenetic Analyses

We used MEGA7 to construct a maximum likelihood (ML) phylogenetic tree of the HML-9 proviruses and solo LTRs to confirm the categorization of the elements identified. The HML-9 proviruses and solo LTRs were compared with the reference sequence of LTR14C-HERVK14C-LTR14C. The proviral sequences with length greater than 80% of the reference and the separate elements, including *gag*, *pro*, *pol*, *env*, and solo LTRs with length greater than 90% of the corresponding segments of the HML-9, were screened. Finally, a total of 5 typical proviruses, 15 *gag*, 6 *pro*, 5 *pol*, 14 *env*, and 37 solo LTRs were analyzed to construct the phylogenetic tree. Meanwhile, 44 solo LTRs with length greater than 90%, 5 proviral sequences with length greater than 80% in the human genome, and 10 *gag*, 8 *pro*, 11 *pol*, and 13 *env* elements with length greater than 90% in the corresponding segments of the HML-9 coding region were used to construct the phylogenetic tree. Using the model selection function in MEGA7, the best nucleotide substitution models for solo LTRs and proviruses were calculated as K2 + G and GTR + G + I, respectively, and the most suitable nucleotide substitution models for analyzed *gag*, *pro*, *pol*, and *env* were HKY + G + I, GTR + G + I, GTR + G + I, and GTR + G, respectively. The nearest-neighbor exchange (NNI) procedure was used to search for tree topologies. Bootstrap tests with 500 bootstrap replicates were used to determine the confidence level of each node in the phylogenetic tree, and, finally, the ML tree was drawn using iTOL (https://itol.embl.de/, accessed on 6 February 2024).

### 2.4. Estimated Time of Integration

We used different strategies to estimate the integration time of an element: 2-LTR region-based and internal coding region-based. For the 2-LTR region-based method, the logic is that the two LTRs of the same provirus were identical at the time of integration and that each accumulated mutation independently. Considering the co-evolution of HML-9 elements with the host genome, we used a replacement rate of 0.2%/nucleotide/million years to assess the effect of divergence on each HML-9 element. The integration time was calculated according to the formula T = D/0.2/2, where T represents the estimated integration time (in millions of years) and D is the percentage of divergent nucleotides between 5′ and 3′ LTR of a proviral element. However, due to evolutionary patterns such as deletion, insertion, and rearrangement, many elements lack either or both LTR regions. Therefore, the method based on 2-LTR regions is not suitable for the elements that lack either or both LTR regions. Thus, for the internal coding regions, we also adopted another strategy, using the formula T = D/0.2 to estimate the integrated time of four internal regions, where T represents the estimated integration time (in millions of years) and D is the percentage of divergent nucleotides between each internal element and the consensus generated. The internal regions contain multiple sequence differences due to the mutations accumulated during viral replication cycles, with a much higher error rate. In contrast, the 5′ LTR and 3′ LTR are identical when the provirus is integrated into the host genome. Therefore, the differences inevitably lead to LTRs being more accurate timing starting points for integration time estimation. For this reason, we have put the integration time calculated based on the 2-LTR as the primary information in the current work.

### 2.5. Conserved Motif Analysis

Conserved motifs of identified HML-9 elements were searched for using the Multiple Em for Motif Elicitation (MEME) Version 5.4.1 online tool (https://meme-suite.org/meme/tools/meme, accessed on 18 August 2022). The following parameter settings were used: motif discovery mode: classic mode; sequence alphabet: DNA, RNA, or protein; site distribution: zero or one occurrence per sequence (zoops); and 10 motifs. The discover program was run iteratively until the 10 most significant motifs were found. Subsequently, the relevant mast files were downloaded, and the final MEME results were visualized using TBtoolsV1.098 software.

### 2.6. Functional Prediction of cis-Regulatory Regions and Enrichment Analysis

In this study, we conducted functional prediction and enrichment analysis of cis-regulatory regions of HML-9 proviral LTRs and solo LTRs to establish the biological significance. We performed genomic region enrichment using the Genomic Regions Enrichment of Annotations Tool (GREAT) (http://great.stanford.edu/public/html/, accessed on 10 June 2022) to analyze gene annotations near LTRs. The association rules were base + extension: 5000 bp upstream, 1000 bp downstream, and a maximum extension of 1,000,000 bp, including selected regulatory structural domains. After identifying potential regulatory genes, the functional enrichment was analyzed using a web-based gene set analysis tool (WebGestalt) (http://www.webgestalt.org/, accessed on 29 June 2022). The enrichment method used in the current work is overrepresentation analysis (ORA). The parameters of the enrichment analysis are as follows: minimum number of IDs in a category: 5; maximum number of IDs in a category: 2000; FDR method: Benjamini–Hochberg (BH); and significance level: top 10.

### 2.7. PBS Type

MEGA7 and BioEdit were utilized to analyze the PBS sequences extracted from HML-9 proviruses as well as the HML-9 reference sequence. These sequences were aligned using tRNAdb (http://trna.bioinf.uni-leipzig.de/, accessed on 20 November 2023) to identify tRNAs. The overall conservation of the PBS sequence across the HML-9 proviruses was depicted using a logo generated by WebLogo (http://weblogo.berkeley.edu, accessed on 20 November 2023), based on the nucleotide alignment of all the HML-9 PBS sequences, in which the letter height was proportional to the nucleotide conservation at each position.

## 3. Results

### 3.1. HML-9 Element Identification, Localization, and Distribution in Chimpanzee Genome Assembly January 2018 (Clint_PTRv2/panTro6)

We mined the sequences of the latest chimpanzee genome assembly January 2018 (Clint_PTRv2/panTro6) using the UCSC Genome Browser Network. Then, using the LTR14C-HERVK14C-LTR14C consensus sequences assembled from DFAM datasets as a query, and comparing these coordinates and sequences with those obtained via searching with the BLAT tool, we identified a total of 26 ERV-K HML-9 provirus elements. Each ERV-K HML-9 element is named according to the genomic locus of integration, utilizing a previously proposed type of nomenclature for HERV-K [49] (Table 1). Element length analysis showed that 6 elements of the ERV-K HML-9 provirus were longer than 70% of the reference length, 8 elements were between 40% and 70% of the reference length, and the remaining 12 elements were below 40% of the reference length. There were 13 loci deletions in 26 ERV-K HML-9 proviruses; specifically, 2 loci were deletions/insertions (chrY: 16516529-16520265; chrY: 10496235-10498083). In addition, 38 solo LTR elements of HML-9 were characterized (Table 2). Of these, 37 solo LTRs were longer than 90% of the representative reference LTR14C, with one deletion/insertion site (chr14: 19027337-19028300). The nucleotide sequence of each element is shown in Appendix A. The overall distribution of CERV-K HML-9 elements indicates that HML-9 was not detected on chromosomes 9, 17, 20, and 22 (Figure 1A), and HERV-K HML-9 was not detected on chromosomes 9 and 22 (Figure 1B). These sequences were not evenly distributed across the chimpanzee and human genomes.

Next, the expected integration number of HML-9 elements per chromosome was predicted and compared with the actual quantity of loci detected to evaluate the actual insertion rate of HML-9 in the chimpanzee genome. The number of HML-9 integration events observed is usually inconsistent with the expected number. For provirus elements, the number of HML-9 insertions on chromosomes 8, 15, 16, 19, and Y was higher than expected. In particular, the frequency of HML-9 provirus element-related sequences on the Y chromosome was very high, with the actual number of integrated provirus elements being 15, which statistical analysis showed to be significantly higher than the value predicted using the chi-square test (*p* < 0.05). On chromosomes 4, 5, 6, and 7, the actual number of identifications was lower than anticipated (Figure 1C). Notably, HML-9 provirus integration was not detected on chromosomes 1, 2B, 3, 9, 10, 11, 12, 14, 17, 18, 20, 21, 22, or X. For solo LTRs elements, the number of HML-9 solo LTRs on chromosomes 2A, 2B, 3, 7, 14, 21, X, and Y was higher than the expected number of integrations. On chromosomes 1, 4, 8, 10, 11, and 12, the actual number identified was lower than expected. The integration of solo LTRs is depicted in Figure 1D. Regarding the proviral elements in the human genome, the number of HERV-K HML-9 insertions on chromosomes 8, 13, 15, 16, 19, 21, and Y was higher than expected. On chromosomes 1, 2, 4, 5, 6, 7, 10, and 12, the actual numbers identified were lower than expected with respect to the solo LTR elements. However, the number of HERV-K HML-9 solo LTRs on chromosomes 2, 3, 14, 15, 18, 21, X, and Y was higher than expected. On chromosomes 1, 4, 5, 6, 7, 8, 10, 11, 12, 13, 17, and 20, the actual numbers identified were lower than expected [46]. The results showed that the distribution of HML-9 provirus and solo-LTR in the chimpanzee chromosomes was non-random.

In addition, all 26 identified proviral elements and 38 solo LTR elements were analyzed to determine their locations in intergenic regions, introns, or exons. The results showed that 22 HML-9 provirus elements were located in the intergenic region, accounting for 84.62% of all proviral elements. Two proviral elements were located in introns (chr5: 60565330-60572694; chr8: 76757424-76759791), one in exons and introns (chr15: 23660410-23668998), and one in genic or intergenic (chrY: 16516529-16520265) (Table 1). In the human genome, the results showed that 13 proviral elements were located in the spacer region, accounting for 56.52% of the total number of proviral elements. Overall, four proviral elements were located in introns, five proviral elements were located in both introns and exons, and one in exons or intergenic [46]. Of the HML-9 solo LTRs in the chimpanzee genome, 30 (78.95%) were located in the intergenic region, and the remaining 8 (21.05%) were located in the intron (Table 2). Of the HML-9 solo LTRs in the human genome, 28 (59.57%) were located in intergenic regions, and the remaining 19 (40.43%) were located in introns [46]. Previous studies have indicated that ERVs accumulate in the human germ line in a two-step process: the integration of targeting bias to guide initial accumulation is followed by purification selection, leading to a loss of provirus, thus disrupting gene function. In particular, the accumulation of HML-2 proviruses in the intergenic regions and intron is targeted during the selection of proviruses integrated into exons and gene regions [50]. As the closest living relative of humans, the similarities and differences of HML-9 element distribution between chimpanzee and human is critical to understanding the co-evolution of HML-9 and the host. Here, we added the comparison of chimpanzee HML-9 and their counterparts in the human genome, including both the provirus (Table 3) and solo LTR forms (Table 4). From the two tables, we can see that there are 38 homologous ERV elements that have been detected in both genomes, including 10 proviral elements and 28 solo LTRs. There are 26 ERV elements (16 proviral elements and 10 solo LTRs) that were only detected in the chimpanzee genome. And there are 32 ERV elements (13 proviral elements and 19 solo LTRs) that were only detected in human genomes.

### 3.2. Structural Characterization of HML-9 Proviral Sequences

To characterize the structure of HML-9 elements, 26 proviruses were further analyzed via comparison with the reference LTR14C-HERVK 14C-LTR14C. According to the annotation information summarized in the Dfam database (https://www.dfam.org/family/DF0000193/features, accessed on 27 January 2021), the complete HML-9 reference presents a typical provirus structure containing four open reading frames (ORFs) and two flanking LTRs. Specifically, the 5′ LTR is located between nucleotides 1 and 587, the CDS of the HERVK14C gag protein ranges from nucleotides 758 to 2548, the CDS of the HERVK14C proprotein covers nucleotides 2548 to 3435, the CDS of the HERVK14C pol protein ranges from nucleotides 3411 to 6060, the CDS of the HERVK14C env protein ranges from nucleotides 5975 to 8020, and 3′ LTR spans from nucleotides 8022 to 8608. We aligned 26 ERV-K HML-9 provirus sequences and annotated the locations and deletions of the individual retroviral components to describe the structure of each HML-9 provirus element (Figure 2). Table 5 summarizes the completeness of six different regions (5′ LTR, *gag*, *pro*, *pol*, *env,* and 3′ LTR). Based on our analysis, it was discovered that a significant portion of CERV-K HML-9 members possess flawed proviral structures. Among these structurally incomplete sequences, seven stood out as particularly defective, measuring less than 2000 bp in length (Figure 2). In stark contrast to the DFAM reference, only six sequences had proviral structures that were greater than 70% of the length of the complete reference sequence (chr 16: 19355716–19364289, chr 2A: 82212507–82221098, chr 8: 43516710–43525262, chr 15: 23660410-23668998, chr 13: 66107993-66115813, and chr 5: 60565330-60572694 exceeded 70% of the complete reference sequence in length). In the human genome, HML-9 16p12.3, 2p12, 15q21.1, 8p11.1, 13q31.1, and 4q33 are longer than 70% of the complete reference sequence in length, and only one sequence measures less than 2000 bp in length.

### 3.3. Phylogenetic Analyses

To gain greater insight into the evolutionary relationships of the identified ERV-K HML-9 elements, including the screened typical provirus sequences, structural regions (*gag*, *pro*, *pol*, *env*), and solo LTRs, we generated phylogenetic ML trees, in which DFAM HERV-K group (HML-1–10) sequences were used as references and the exogenous betaretrovirus mouse mammary tumor virus (MMTV), Mason–Pfizer monkey virus (MPMV), and Jaagsiekte sheep retrovirus (JSRV) sequences were used as outgroups. All five CERV-K HML-9 proviruses and five HERV-K HML-9 proviruses (greater than 80% of the ERV-K HML-9 reference sequence length) were clustered with the Dfam ERV-K HML-9 reference sequence with 100% bootstrap support (Figure 3A). Next, ML trees with a total of 37 CERV-K and 44 HERV-K HML-9 solo LTRs (greater than 90% of the length of the LTR14C reference sequence) were constructed together with the LTR14C reference sequence (Figure 3B). In addition, ML trees were created for four sub-region trees (greater than 90% of the corresponding part of the reference sequence), including 15 CERV-K and 10 HERV-K *gag* elements, 6 CERV-K and 8 HERV-K *pro* elements, 5 CERV-K and 11 HERV-K *pol* elements, and 14 CERV-K and 13 HERV-K *env* elements. The results showed that these phylogenetic clusters of different regions of ERV-K HML-9 were all clustered together and distinctly separated from other HERV-K clusters (HML1–8, 10) (Figure 3C–F).

### 3.4. Estimated Time of Integration

A special characteristic of the provirus sequence is that the 5′ LTR and 3′ LTR of the same provirus are identical at the time of integration. Subsequently, they accumulate random substitutions independently, similar to the internal provirus sequences and host genomes, the number of whose nucleotide divergences between LTRs can be used to assess the time of integration of the provirus [51]. The T value was estimated using the relation T = D/0.2/2. However, due to evolutionary pressures such as deletion, insertion, and rearrangement, the lack of one or two LTR proviruses would affect such an estimation. Therefore, we also used the divergence data between a single internal gene part and its consensus to calculate its integration time [52] using the formula T = D/0.2. For each provirus region, we provide detailed information on the time of provirus formation in Table 6. Between the two species—chimpanzee and human—there are only six homologous 2-LTR proviruses at the same locus. We used these six homologous 2-LTR proviruses to calculate the integration time of HML-9 in both human and chimpanzee genomes, and the results showed that the integration time of HML-9 in the chimpanzee genome is 14 mya–27 mya, and that of HML-9 integration in the human genome is 18 mya–28 mya. The average integration time of HML-9 in the chimpanzee genome is 22 mya, and the average integration time of HML-9 in the human genome is 23 mya. The *p*-value was calculated by the unpaired Student’s *t*-test statistical method, and the results showed that there was no statistical difference in the integration age between the two species (*p* = 0.8553). To further confirm the correctness of estimated integrated time based on LTR regions and the calculation method, we compared the two species at an overall level. The time of HML-9 insertion in chimpanzees calculated by the LTR-based method ranged from 14 to 36 mya, with an average integration time of 25 mya, and the time of integration of HML-9 in the human genome calculated by the LTR-based method ranged from 18 to 49 mya, with an average integration time of 29 mya. The *p*-value was calculated by the unpaired Student’s *t*-test statistical method, and the results showed that there was also no statistical difference in the integration time between the two species, suggesting once again the similar ages of the same set of proviruses in humans and chimpanzees. The minor differences of HML-9 integration into human and chimpanzee genomes can be explained as follows: the two species have undergone independent evolutions of 6.5–7.5 mya since their divergence, and currently there are only 3 homologous proviruses between the two species. The remainders are non-homologous proviruses, suggesting different shaping abilities towards these foreign elements from different hosts after divergence. These evolutionary differences may contribute to the differences in integration times. These results also prove the effectiveness of the calculation method. The divergence between human and chimpanzee ancestors is indicated to trace back to approximately 6.5–7.5 mya or earlier. Our results here revealed that the chimpanzee-specific insertion time was indeed similar to that of human-specific insertion and confirmed that HML-9 was integrated into common ancestors before humans and chimpanzees diverged.

However, most of the ERV-K HML-9 elements (*gag*, *pro*, *pol*, and *env*) found in the chimpanzee genome are integrated between 22 to 118 mya, with an average integration time of 45 mya. There exists a very large discrepancy between the two methods. A reasonable explanation for the difference between the two methods is as follows: The 5′ LTR and 3′ LTR are identical when the provirus is integrated into the host genome. However, the internal regions contain multiple sequence differences due to the mutations accumulated during viral replication cycles, with a much higher error rate. The difference inevitably leads to LTRs being more accurate timing starting points for integration time estimation, suggesting the use of LTR analysis as the primary method of age estimation. To summarize, our article uses the integration time calculated based on the LTR as the primary information.

### 3.5. Conserved Motif Analysis

For DNA, the motif refers to the DNA binding site, which is a small DNA sequence to which transcription factors or other transcription-related proteins can bind to regulate transcription. MEME Suite is an online website for the prediction, assembly, and annotation of motif tools to identify conserved motifs in CERV DNA sequences. MEME tools were used to perform prediction and analysis. Appendix A shows the top 10 conserved MEME motifs in CERV DNA sequences. The motifs of each sequence were visualized, with different color squares representing different motifs and the size of the squares representing the length of the motifs. This information allowed us to visualize the similarities and differences between the motifs of different sequences. Appendix A shows the specific structure of the motif and the expected value (*p*-value) of the motif significance. In the model, the site value is the number of sequences containing the motif found, and the width is the specific sequence length of the motif.

### 3.6. Functional Prediction of cis-Regulatory Regions and Enrichment Analysis

GREAT is a tool for the gene annotation of peak regions. After obtaining the gene annotation information, we wanted to further obtain the functional information of the gene. WebGestalt constitutes a set of widely used gene enrichment analysis tools for functional enrichment analysis in the background of different organisms. It is a powerful, integrated data mining system, which can manage, retrieve, organize, visualize, and statistically analyze many genes. To further understand the biological role of these loci, we used GREAT and WebGestalt to annotate genes and subsequently performed gene ontology (GO) annotation and Kyoto Encyclopedia of Genes and Genomes (KEGG) pathway enrichment analysis. It is worth noting that there is no chimpanzee reference genome in the GREAT tool and that this can be converted into a supported combination using the UCSC LiftOver utility. That is, LiftOver is used as the best approximation for mapping regions in the chimpanzee genome to homologous regions in the human genome. The first 10 enrichment pathways of GO and KEGG are shown in this study.

For the CERV-K HML-9 provirus LTR, a total of 37 genes were predicted. Of these, 2 proviral LTRs were not associated with any gene, 3 proviral LTRs were associated with one gene, and 17 proviral LTRs were associated with two genes (Appendix A; Figure 4A). The absolute distance from no gene to the transcription initiation site (TSS) is less than 5 kb. All told, 16 genes had an absolute distance to the TSS of between 5 and 50 kb; 13 genes had an absolute distance to the TSS of between 50 and 500 kb; and 8 genes had an absolute distance to the TSS of more than 500 kb (Figure 4B,C). We also generated GO slim summaries for the biological classification of the key genes associated with the proviral LTR. The GO analysis of biological process (BP) summary showed these genes to be mainly enriched due to metabolic process, response to stimulus, and cellular component organization. The GO analysis of cellular components (CC) summaries indicates that these genes are significantly enriched in the membrane, endomembrane system, and nucleus, while the GO analysis of molecular function (MF) summaries indicates that these genes are significantly enriched in protein binding, ion binding, and transferase activity (Figure 4D–F). In addition, these potential regulatory genes were annotated according to selected functional categories and enriched for analysis. The three most important terms for BP according to FDR values are epoxygenase P450 pathway, exogenous drug catabolic process, and arachidonic acid metabolic process (Figure 5A). The volcano plots in Figure 5B,D,F show the log2 of the enrichment of FDR relative to all functional categories in the database, highlighting the degree of separation of important categories from the background. The size and color of the dots are proportional to the amount of overlap (for ORA). Significantly enriched categories are tagged, and labels are automatically located using a force-field-based algorithm at startup. According to the FDR values, the three most important terms for CC are U2 snRNP, integral components of postsynaptic density membrane, and intrinsic components of postsynaptic density membrane (Figure 5C). Additionally, the three most important terms for MF are nucleoside transmembrane transporter activity, arachidonic acid monooxygenase activity, and arachidonic acid epoxygenase activity (Figure 5E).

We also performed the above analysis for CERV-K HML-9 solo LTRs and all CERV-K HML-9 elements, which we will not repeat here in detail, as shown in the Appendix A. It is important to note that these results are based solely on predictions, and that many factors in the prediction process can affect the accuracy of the process. Further investigation is needed to confirm any implied associations between individual LTRs and nearby genes.

### 3.7. PBS Type

Hosts often target the relatively conserved regions in rapidly mutating retroviruses to inhibit their replication. One of these regions is called PBS, which must be complementary to the host tRNA to initiate reverse transcription. Among the 26 ERV-K HML-9 proviruses analyzed, 9 conserved a PBS sequence. As expected, when present, the PBS sequence is located three residues downstream of the 5′ LTR and is 18 nucleotides in length. Eight out of nine analyzed PBS were predicted to recognize a lysine (K) tRNA and show a conserved nucleotide composition, as indicated in the logo generated from the PBS sequence alignment (Appendix A, Figure 6). In the human genome, three homologous HML-9 PBS sequences were predicted to recognize lysine (K) tRNA.

## 4. Discussion

The HML-9 elements in the chimpanzee genome were identified via bioinformatic assessment. The performance was in line with the types of studies presented for the other HML groups [47,48,53]. In the present study, we identified and characterized 26 proviruses and 38 solo LTRs retrieved in the chimpanzee genome assembly January 2018. Additionally, we analyzed the distribution, structural characterization, phylogeny, integration time, motifs, and PBS sequence features of all ERV-K HML-9 elements in detail, predicting regulatory function, in order provide a comprehensive characterization of the ERV-K HML-9 group.

Transposable elements (TEs) are DNA sequences that can be moved or replicated in the genome and integrated into new sites within it, contributing nearly half of the open chromatin region of the human genome and elements unique to most primates [54,55,56,57,58,59]. Based on whether the transposition process forms RNA intermediates, transposons can be classified as DNA transposons and retrotransposons. Retrotransposons are RNA-mediated, being accompanied by a retro-transcription process that produces a new copy at a new location in the genome in a copy-and-paste fashion [60]. In contrast, the transposon mechanism of DNA transposons is in a cut-and-paste format [61]. ERVs are related to LTR retrotransposons. There are two generally accepted models for the mechanism of ERV proliferation in the host genome: the evolutionary model of the reinfection of germ cell lines and the retrotransposon model [62]. However, ERVs can be reactivated by a variety of factors, including infectious agents, exogenous viruses, radiation, aging-related processes, epigenetic drugs, cytokines, and mitogens [63]. Studies have identified replication-competent ERVs in many species, and recombination between defective ERVs may also lead to the production of infectious viruses [64,65]. Abnormally activated ERVs may be involved in the occurrence and development of tumors [66,67,68,69].

All ERV families discovered in humans have subsequently been found in other primates, although some of the younger HERV genes are not conserved in other species [45]. Here, we have added the comparison of chimpanzee HML-9 and their counterparts in the human genome in Table 3 and Table 4, including both the full and single LTR forms. In total, there are 38 ERV elements that have been detected in both genomes, including 10 proviral elements and 28 solo LTRs. There are 13 proviral elements and 19 solo LTRs that have only been detected in human genomes. And there are 16 proviral elements and 10 solo LTRs that have only been detected in the chimpanzee genome. The divergence between human and chimpanzee ancestors is known to trace back to approximately 6.5–7.5 mya or earlier. Our results confirmed that HML-9 was integrated into common ancestors before humans and chimpanzees diverged. Therefore, in theory, the chimpanzee-specific insertion should indeed be the same as the human-specific insertion. However, the two species have undergone independent evolutions of 6.5–7.5 mya since their divergence. The differences in HML-9 element distribution between the two species displayed here exactly validate their independent and separate evolution.

CERVs contribute to species-specific genomic changes in the chimpanzee genome due to the differentiation between chimpanzees and humans, potentially causing the source of the genomic differences between chimpanzees and their closest relatives, humans. The function of ERVs is varied, owing to the genomic structure of the integration site and the modification of the proviral sequences. The induction of novel transcriptional activity in the genomic region of the integration site can trigger cancer and abnormal developmental processes. The ERV provirus has attracted increasing interest because of its association with cancer, autoimmune diseases, and neurodegenerative disorders. Studies have reported clear signals of TE-derived, tissue-specific putative enhancers, as well as promoters that are specific to humans or chimpanzees. In particular, research has identified LTR5 as a putative promoter in induced pluripotent stem cells (iPSC) [14], while in other studies it has been found to exhibit enhancer activity in human NCCIT cells [70]. LTR5_Hs/LTR5 showed higher expression levels, displaying high activity in embryonic stem cells (ESC) and extended pluripotent stem cells (EPSC). LTR5_Hs/LTR5 can act as a distal enhancer of regulatory host genes [11]. Among the class II elements, the HERV-K sequence was initially identified due to its similarity to the mouse breast tumor virus (MMTV) and was classified into 10 so-called human MMTV-like branches (HML1–10) [7,71]. Of these, HML-2 has the latest time integration and is the most bioactive, containing the youngest known HERV sequence in addition to many members of the full-length ORF. Furthermore, it is the only known element with human-specific integration [72,73]. HML-2 was first integrated into the genome of the common ancestor of humans and Old World monkeys 30 million years ago, and it contains more than 12 elements that have been integrated since the divergence of humans and chimpanzees [74,75,76]. In particular, a study identified two species-specific HERV-K (HML-2) provirus loci (Pan8q and Pan2Ap) in chimpanzees, with Pan2Ap represented as a solo LTR in the reference genome and showing a dimorphism between solo LTR and provirus [77]. HERV-K solo LTR, formed after the differentiation of humans and chimpanzees, has been identified in the chimpanzee genome. As a substitute promoter and enhancer, solo LTRs play significant roles in gene regulation. They are thought to contribute to species evolution by regulating host gene networks and key host genes, especially those involved in embryogenesis and stem cell development [58,70,78]. Studies have shown that chimpanzee-specific ERVs located in the genomic region between chimpanzee PNRC2 5′ UTR1 and 2 can induce alternative splicing or different RNA polymerase II binding sites on the genes, indicating that chimpanzee-specific ERVs can generate alternative transcripts through their new insertion in the genes [40]. LTRs of primate-specific retrovirus MER41 function as natural enhancers of interferon-inducible gene networks [79]. Previous studies into HML-9 indicate that ERV-K HML-9 can function in different tissues under physiological conditions and during disease progression, which may contribute to immune regulation and antiviral defense [80]. Comprehensive information has been reported regarding ERV-K HML-9 in the human genome. However, due to the lack of a comprehensive description of the ERV-K HML-9 group in the chimpanzee genome, the specific contribution of a single ERV-K HML-9 locus to the chimpanzee transcriptome remains unclear. In this study, we provide in great detail the complete features of all 64 ERV-K HML-9 elements retrieved from chimpanzee genome assembly January 2018.

Firstly, we predicted the expected integration number of ML-9 elements per chromosome and compared the result with the actual number of loci detected to evaluate the actual insertion rate of HML-9 in the chimpanzee genome. The number of ERV-K HML-9 integration events observed is usually not consistent with the expected number. The results showed that the distribution of these provirus and solo LTRs showed a non-random integration pattern, and these elements were mainly distributed in intergenic regions and introns. In particular, the number of proviruses on the Y chromosome was significantly different from that predicted by the chi-square test, indicating that the Y chromosome accumulated a higher density of CERVs and their related sequences. Such a pattern of distribution is in agreement with those results observed in the ERV-K HML-9 group’s human genome in general [46]. The initial integration on the Y chromosome may be disfavored due to its heterochromatic status; however, once proviruses are integrated, they tend to have minimal detrimental effects since the Y chromosome is gene-sparse. In addition, because Y lacks a homolog, newly-integrated DNA cannot be removed through homologous recombination, resulting in the accumulation of a large number of complex repetitive sequences on the Y chromosome [81]. The comparison between humans and chimpanzees also shows that HML-9 elements are not distributed on chromosomes 9, 17, 20, and 22 in the chimpanzee genome, while there is an absence of HML-9 elements only on chromosomes 9 and 22 in the human genome, suggesting that chimpanzees lost two homologous HML-9 elements during evolution.

Secondly, we sought to define the structural features of the chimpanzee genome CERV-K HML-9 provirus type relative to the consensus annotation of all insertions and deletions in the internal sequence. The characterization of the HML-9 consensus sequence confirmed a structure resembling the typical proviral genome, with the retroviral genes *gag*, *pro*, *pol*, and *env* flanked by 5′ LTRs and 3′ LTRs. The results show that only six elements (23.08%) maintain a relatively complete structure, while most of the sequences leave the genome structure incomplete due to deletions, with seven being less than 2000 bp in length (26.92%). We also annotate all the minor insertions and deletions, which can provide a specific background for the study of the structure of a single HML-9 locus. Six elements in the human genome (26.09%) maintain a relatively intact structure, and only one element is less than 2000 bp in length (4.35%). The results suggest that the absence of ERV-K HML-9 elements is more pronounced in the chimpanzee genome compared to the human genome. In the genomes of chimpanzees and humans, there are identical pol deletions in some proviruses, most of which are on the Y chromosome. We examined the HML-9 proviral DNA sequence of the Y chromosome in the chimpanzee genome using MEGA7 and found that the flanking of the integration site transcribed from the same chain had exactly the same DNA sequence (Appendix A).

Next, phylogenetic analysis showed that homologous HML-9 elements in the human and chimpanzee genomes are clustered together. However, the CERV-K HML-9 elements sequence had no obvious clustering and formed a single phylogenetic group. This was significantly different from other HML groups. Then, the integration time of ERV-K HML-9 provirus was calculated using the regions of LTRs, namely, *gag*, *pro*, *pol*, and *env*. The results show that LTR integration ranged from 14 to 36 mya, with an average integration time of 25 mya. However, the major cycle of ERV-K HML-9 integration based on four internal regions was between 22 and 118 mya, with an average integration time of 45 mya. Overall, the integration time estimated using LTR elements was later than the times estimated based on the four regions (*gag*, *pro*, *pol*, and *env*). The difference in estimated integration times between these two methods may be due to internal coding regions accumulating mutations during each replication cycle, resulting in the internal regions containing multiple sequence differences with a much higher error rate, while two identical LTRs integrate into the host genome during the integration phase. Therefore, it is more reasonable to use the integration time of LTRs to evaluate the integration time. In the human genome, LTRs integrated between 18 and 49 mya, with an average integration time of 29 mya, which was earlier than the integration time in the chimpanzee genome. Based on the integration time estimated by the LTR method, the time difference of HML-9 integration into human and chimpanzee genomes is 4 mya, which can be explained as follows: ① Between the two species—human and chimpanzee—there are only three homologous proviruses, most of which are non-homologous provirus, which may also contribute to the differences in integration times. ② Some are located in regions differentially subjected to other postintegration rearrangement (segmental duplication or deletion).

In addition, we performed motif-conserved analysis of chimpanzee gene family DNA sequences and obtained the top ten motifs with the highest frequency in proviruses, solo LTRs, and two LTR proviruses. Immediately after, we predicted and clustered the potential regulatory genes of ERV-K HML-9 provirus and solo LTRs. For the ERV-K HML-9 provirus, a total of 37 genes were predicted. Analysis shows that these genes are related to biological regulation. Previous studies have shown that among the six H3K9 methyltransferases present in mammals, SETDB1 has a specific and nonredundant role in the deposition of H3K9me3 in compartment A, where it is tethered by hundreds of sequence-specific KRAB domain-containing zinc finger transcription factors, primarily to repress endogenous retroviruses [82,83,84]. In conclusion, it is important to identify specific ERVs associated with certain diseases, especially ERVs polymorphic loci, which may influence the expression profile of viruses in different individuals and the regulation of host genes. For ERV-K HML-9 solo LTRs, a total of 54 genes were predicted. Analysis showed that these genes were related to synapses. Previous studies showed that HERV produces proteins that regulate brain cell function and synaptic transmission and which are implicated in the etiology of neurological and neurodevelopmental psychiatric disorders, and investigators combined single-molecule tracking, calcium imaging, and behavioral approaches to demonstrate that the envelope protein (Env) of HERV-W is usually silent but can be expressed in patients with neuropsychiatric disorders, altering N-methyl-d-aspartate receptor (NMDAR)-mediated synaptic organization and plasticity through glial- and cytokine-dependent changes [85]. It must be noted that these results are based entirely on predictions, the accuracy of which is determined by many factors, and further investigation is needed to confirm any implied associations between individual LTRs and nearby genes.

During the initiation phase of retroviral replication, host tRNAs, adopted as a primer for retroviral reverse transcriptase, are partially unfolded from their native structure to facilitate the PBS being base-paired to a specific complementary sequence on the viral genomic RNA. For the PBS analysis of the chimpanzee ERV-K HML-9 elements, the results showed that the TGG initiation nucleotide was the most conserved among the 18 bases. This result also applies to the human genome. We identified nine proviral PBS sequences, and three of them belonged to lysine. The logo maps generated from the human genome and chimpanzee genomes were identical. In the human genome, eight homologous HML-9 PBS sequences were predicted to recognize lysine (K) tRNA. It should be noted that these results are entirely based on prediction. Experimental verification studies are needed to confirm the association between these elements and these genes.

It is worth noting that in this study we are lacking data related to conserved transcription factor binding sites due to the absence of chimpanzee classification in the Species section of JASPAR.

## 5. Conclusions

The identification, characterization, and comparative genomics of CERVs introduced in this report will not only contribute to our understanding of the functional significance of these elements in chimpanzees but also contribute to a better understanding of the role of endogenous retroviruses in primate evolution. We believe that further examination of other ERV families will add more hitherto unknown aspects of ERVs, thereby providing more information about the overall biological role of endogenous retroviruses. The biological function of ERVs and their association with the disease is largely elusive. More research is needed to understand the role of ERVs in various diseases and elucidate its role in the pathogenesis of diseases, providing new ideas for the further discovery of disease-associated antigens and the development of new treatments for diseases.

## Figures and Tables

**Figure 1 viruses-16-00892-f001:**
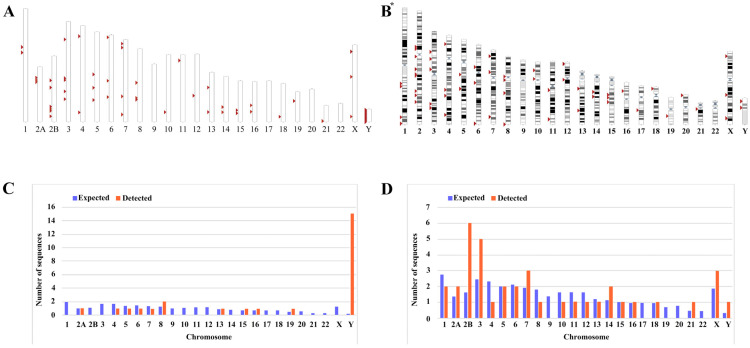
Chromosomal distribution of ERV-K HML-9 loci in chimpanzee. (**A**) All BLAT-identified HML-9 elements (red arrows) are displayed on the chimpanzee karyotype. (**B**) All HERV-K HML-9 elements (red arrows) are displayed on the human karyotype. The number of HML-9 proviral elements (**C**) and solo LTRs (**D**) integrated into each chimpanzee chromosome was determined and compared to the expected number of insertion events. The expected number of sequences in each chromosome is marked in blue, and the actual number of sequences detected is marked in orange. The ERV-K HML-9 proviral enrichment in chromosome Y is particularly clear. The ERV-K HML-9 solo LTR enrichment in chromosomes 2B, 3, 7, 14, and X is particularly clear. * From references [46].

**Figure 2 viruses-16-00892-f002:**
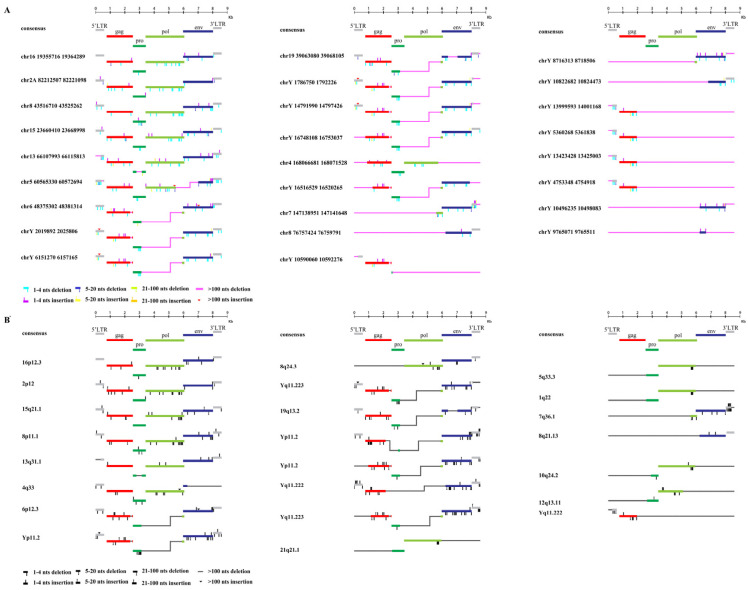
Structural characterization of HML-9 provirus. (**A**) chimpanzee and (**B**) human. Nucleotide insertions and deletions of each ERV-K HML-9 provirus nucleotide sequence are annotated by comparison to the ERV-K HML-9 consensus sequence from Dfam. * From references [46].

**Figure 3 viruses-16-00892-f003:**
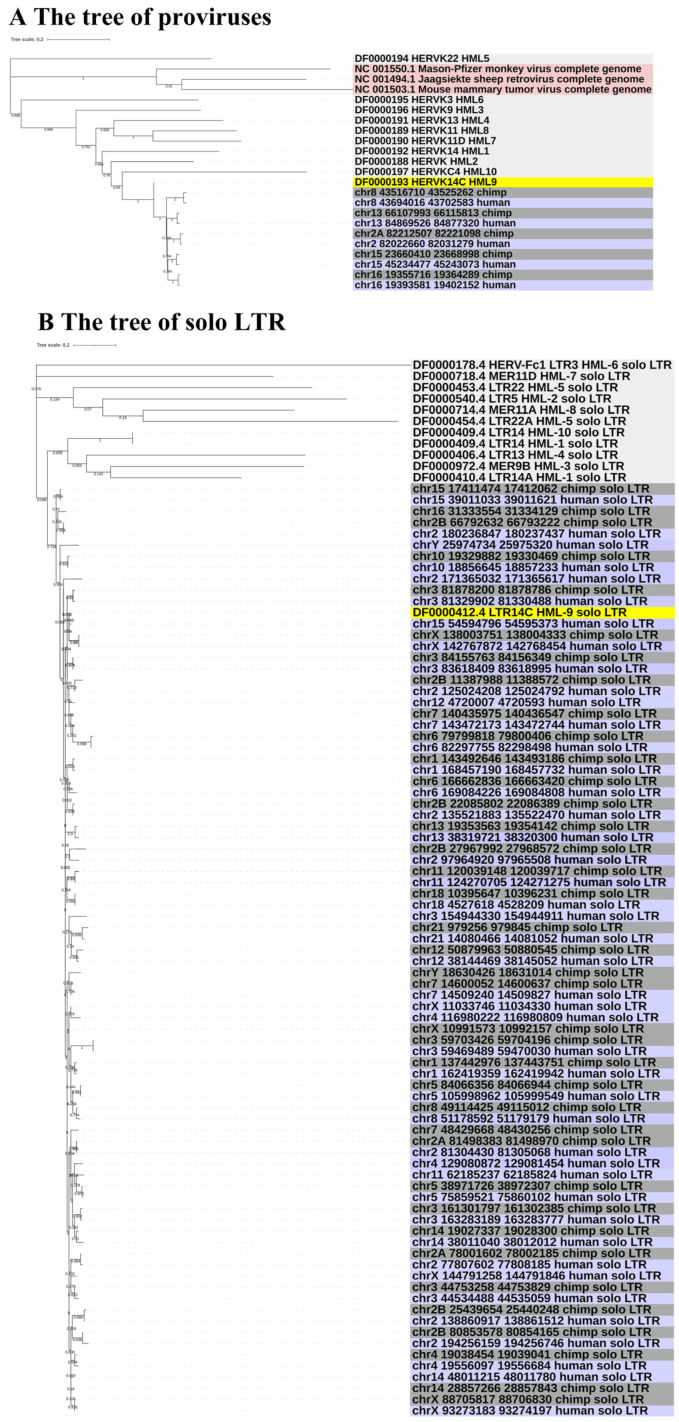
(**A**,**B**) Phylogenetic analysis of the HML-9 near-full-length proviruses and solo LTRs by the maximum likelihood method, including HML-9 elements identified in both chimpanzee and human. Phylogenetic analyses of HML-9 proviral elements identified in both chimpanzee and human (**A**), solo LTRs identified in both chimpanzee and human (**B**), together with references. The resulting phylogeny was tested by the bootstrap method with 500 replicates. The branch length indicates the number of substitutions per site. (**C**–**F**) Phylogenetic analysis of four subregions of the ERV-K HML-9 by the maximum likelihood method, including HML-9 elements identified in both chimpanzee and human. Phylogenetic analyses of ERV-K HML-9 *gag* elements identified in both chimpanzee and human (**C**), *pro* elements identified in both chimpanzee and human (**D**), *pol* elements identified in both chimpanzee and human (**E**), and *env* elements identified in both chimpanzee and human (**F**), together with references. The resulting phylogeny was tested by the bootstrap method with 500 replicates. The branch length indicates the number of substitutions per site.

**Figure 4 viruses-16-00892-f004:**
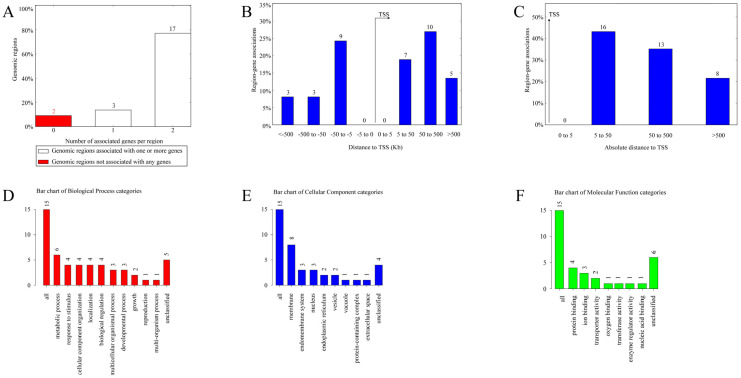
The genes associated with CERV-K HML-9 proviral LTRs and GO slim summaries. (**A**) The number of associated genes per proviral LTR. (**B**) Binned by orientation and distance to TSS. (**C**) Binned by absolute distance to TSS. Biological process (**D**), cellular component (**E**), and molecular function (**F**) summaries are represented by red, blue, and green bars, respectively. The height of the bar represents the number of IDs in the gene list and in the category.

**Figure 5 viruses-16-00892-f005:**
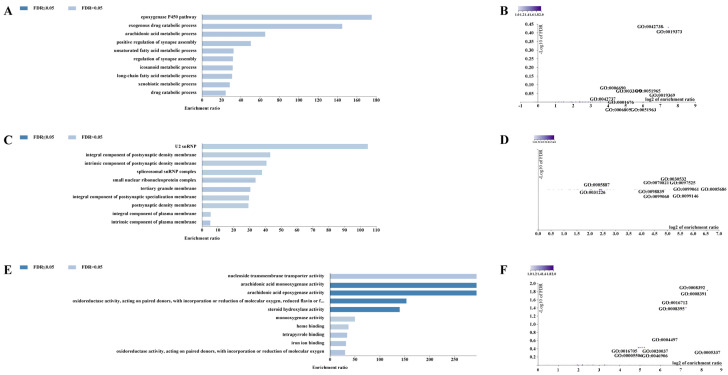
Enrichment result categories binned by biological process, cellular component, and molecular function. (**A**,**B**) Bar chart and customizable volcano plot of the biological process enrichment results. (**C**,**D**) Bar chart and customizable volcano plot of the cellular component enrichment results. (**E**,**F**) Bar chart and customizable volcano plot of molecular function enrichment results.

**Figure 6 viruses-16-00892-f006:**
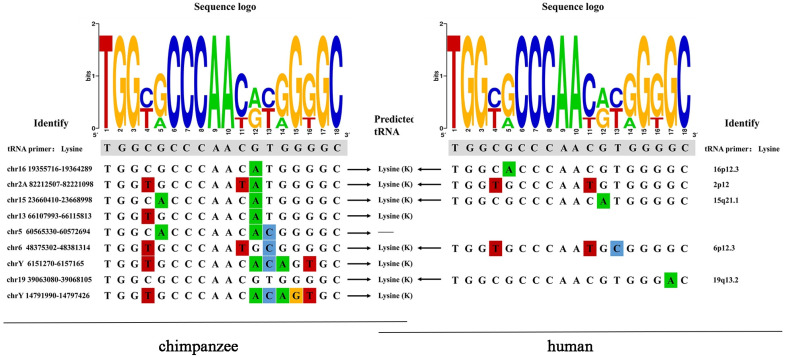
ERV-K HML-9 proviruses of chimpanzee PBS analyses. Nucleotide alignment of the PBS sequences identified in the HML-9 proviruses. In the upper part, a logo represents the general HML-9 PBS consensus sequence: for each nucleotide, the letter height is proportional to the degree of conservation among HML-9 members. Red indicates the T base, green indicates the A base, blue indicates the C base, and orange indicates the G base, and different colors are used to distinguish the 4 bases.

**Table 1 viruses-16-00892-t001:** HML-9 provirus distribution in chimpanzee genome assembly January 2018 (Clint_PTRv2/panTro6).

Number	Chromosome	Strand	Position Start	Position End	Length (bp)	Match + Mismatch (bp)/Full Length (bp)	Range	Q Gap Bases/(Match + Mismatch + Q Gap Bases)	Insertion or Deletion	Intergenic/Intron/Exon
1	chr16	-	19,355,716	19,364,289	8574	95.90%	(90–100%)	1.14%	NA	intergenic
2	chr2A	+	82,212,507	82,221,098	8592	95.75%	(90–100%)	1.32%	NA	intergenic
3	chr8	-	43,516,710	43,525,262	8553	93.95%	(90–100%)	3.39%	NA	intergenic
4	chr15	-	23,660,410	23,668,998	8589	89.68%	(80–90%)	7.89%	NA	exonic&intronic
5	chr13	+	66,107,993	66,115,813	7821	85.42%	(80–90%)	8.35%	NA	intergenic
6	chr5	-	60,565,330	60,572,694	7365	75.51%	(70–80%)	22.56%	deletion	intron
7	chr6	+	48,375,302	48,381,314	6013	64.13%	(60–70%)	35.21%	deletion	intergenic
8	chrY	-	2,019,892	2,025,806	5915	60.86%	(60–70%)	37.99%	deletion	intergenic
9	chrY	+	6,151,270	6,157,165	5896	60.78%	(60–70%)	37.88%	deletion	intergenic
10	chr19	+	39,063,080	39,068,105	5026	56.83%	(50–60%)	41.91%	deletion	intergenic
11	chrY	+	1,786,750	1,792,226	5477	55.32%	(50–60%)	40.62%	deletion	intergenic
12	chrY	+	14,791,990	14,797,426	5437	55.17%	(50–60%)	40.45%	deletion	intergenic
13	chrY	-	16,748,108	16,753,037	4930	54.55%	(50–60%)	39.48%	deletion	intergenic
14	chr4	-	168,066,681	168,071,528	4848	54.03%	(50–60%)	0.96%	NA	intergenic
15	chrY	+	16,516,529	16,520,265	3737	35.85%	(30–40%)	52.96%	deletion;insertion	genic&intergenic
16	chr7	-	147,138,951	147,141,648	2698	27.43%	(20–30%)	11.17%	deletion	intergenic
17	chr8	+	76,757,424	76,759,791	2368	26.57%	(20–30%)	0.48%	NA	intron
18	chrY	+	10,590,060	10,592,276	2217	24.95%	(20–30%)	8.91%	NA	intergenic
19	chrY	+	8,716,313	8,718,506	2194	22.85%	(20–30%)	6.51%	NA	intergenic
20	chrY	-	10,822,682	10,824,473	1792	19.48%	(10–20%)	3.45%	NA	intergenic
21	chrY	+	13,999,593	14,001,168	1576	17.24%	(10–20%)	12.03%	deletion	intergenic
22	chrY	+	5,360,268	5,361,838	1571	17.18%	(10–20%)	12.07%	deletion	intergenic
23	chrY	-	13,423,428	13,425,003	1576	17.03%	(10–20%)	13.10%	deletion	intergenic
24	chrY	-	4,753,348	4,754,918	1571	16.97%	(10–20%)	13.14%	deletion	intergenic
25	chrY	-	10,496,235	10,498,083	1849	6.83%	(0–10%)	67.62%	deletion;insertion	intergenic
26	chrY	-	9,765,071	9,765,511	441	4.72%	(0–10%)	3.10%	NA	intergenic

**Table 2 viruses-16-00892-t002:** HML-9 Solo LTR tracks distribution in chimpanzee genome assembly January 2018 (Clint_PTRv2/panTro6).

Number	Chromosome	Strand	Position Start	Position End	Length (bp)	Percentage of LTR14C in Length	Match + Mismatch/Full Length	Range	Qgap(bp)/(Match + Mismatch + Qgap(bp))	Insertion or Deletion	Intergenic/Intron/Exon
1	chr1	+	137,442,976	137,443,751	776	101.19%	6.90%	(0–10%)	4.81%	NA	intergenic
2	chrX	-	88,705,817	88,706,830	1014	100.85%	6.88%	(0–10%)	2.47%	NA	intergenic
3	chr2B	+	66,792,632	66,793,222	591	100.34%	6.84%	(0–10%)	0.34%	NA	intergenic
4	chr5	-	84,066,356	84,066,944	589	100.00%	6.82%	(0–10%)	0.00%	NA	intergenic
5	chr7	-	48,429,668	48,430,256	589	100.00%	6.82%	(0–10%)	0.34%	NA	intergenic
6	chr15	+	17,411,474	17,412,062	589	100.00%	6.82%	(0–10%)	0.00%	NA	intergenic
7	chr2B	-	80,853,578	80,854,165	588	99.83%	6.81%	(0–10%)	0.00%	NA	intergenic
8	chr3	-	161,301,797	161,302,385	589	99.83%	6.81%	(0–10%)	0.00%	NA	intron
9	chr4	+	19,038,454	19,039,041	588	99.83%	6.81%	(0–10%)	0.17%	NA	intergenic
10	chr8	-	49,114,425	49,115,012	588	99.83%	6.81%	(0–10%)	0.00%	NA	intergenic
11	chr10	-	19,329,882	19,330,469	588	99.83%	6.81%	(0–10%)	0.00%	NA	intergenic
12	chr14	+	19,027,337	19028300	964	99.83%	6.81%	(0–10%)	35.96%	deletion;insertion	intron
13	chr2B	+	22,085,802	22,086,389	588	99.66%	6.80%	(0–10%)	0.17%	NA	intron
14	chr3	-	81,878,200	81,878,786	587	99.66%	6.80%	(0–10%)	0.17%	NA	intergenic
15	chr3	-	84,155,763	84,156,349	587	99.66%	6.80%	(0–10%)	0.17%	NA	intergenic
16	chr6	-	79,799,818	79,800,406	589	99.66%	6.80%	(0–10%)	0.34%	NA	intergenic
17	chr7	+	14,600,052	14,600,637	586	99.49%	6.78%	(0–10%)	0.34%	NA	intron
18	chrY	-	18,630,426	18,631,014	589	99.49%	6.78%	(0–10%)	0.51%	NA	intergenic
19	chr2A	-	81,498,383	81,498,970	588	99.32%	6.77%	(0–10%)	0.51%	NA	intergenic
20	chrX	+	10,991,573	10,992,157	585	99.32%	6.77%	(0–10%)	0.51%	NA	intergenic
21	chr2A	+	78,001,602	78,002,185	584	99.15%	6.76%	(0–10%)	0.68%	NA	intergenic
22	chr18	+	10,395,647	10,396,231	585	99.15%	6.76%	(0–10%)	0.68%	NA	intergenic
23	chrX	-	138,003,751	138,004,333	583	98.98%	6.75%	(0–10%)	0.00%	NA	intergenic
24	chr5	-	38,971,726	38,972,307	582	98.64%	6.73%	(0–10%)	0.00%	NA	intergenic
25	chr12	-	50,879,963	50,880,545	583	98.47%	6.71%	(0–10%)	0.69%	NA	intergenic
26	chr14	+	28,857,266	28,857,843	578	97.96%	6.68%	(0–10%)	2.04%	NA	intergenic
27	chr21	-	979,256	979,845	590	97.79%	6.67%	(0–10%)	2.05%	NA	intron
28	chr6	+	166,662,836	166,663,420	585	97.27%	6.63%	(0–10%)	2.56%	NA	intergenic
29	chr16	-	31,333,554	31,334,129	576	96.76%	6.60%	(0–10%)	2.91%	NA	intergenic
30	chr3	+	44,753,258	44,753,829	572	96.42%	6.58%	(0–10%)	3.41%	NA	intergenic
31	chr2B	+	27,967,992	27,968,572	581	96.25%	6.56%	(0–10%)	3.58%	NA	intron
32	chr7	+	140,435,975	140,436,547	573	96.25%	6.56%	(0–10%)	3.58%	NA	intergenic
33	chr2B	-	25,439,654	25,440,248	595	96.08%	6.55%	(0–10%)	2.76%	NA	intergenic
34	chr13	-	19,353,563	19,354,142	580	95.91%	6.54%	(0–10%)	4.09%	NA	intron
35	chr2B	-	11,387,988	11,388,572	585	95.40%	6.51%	(0–10%)	4.44%	NA	intergenic
36	chr3	-	59,703,426	59704196	771	94.21%	6.42%	(0–10%)	5.63%	NA	intergenic
37	chr11	+	120,039,148	120,039,717	570	94.04%	6.41%	(0–10%)	2.99%	NA	intergenic
38	chr1	+	143,492,646	143,493,186	541	88.07%	6.01%	(0–10%)	3.36%	NA	intron

**Table 3 viruses-16-00892-t003:** The comparison of chimpanzee HML-9 proviral elements and their counterparts in the human genome.

Species	Chromosome	Strand	Position Start	Position End
**chimp ***	**chr2A**	**+**	82,212,507	82,221,098
**human ***	**chr2**	**+**	82,022,660	82,031,279
**chimp**	**chr4**	**-**	168,066,681	168,071,528
**human**	**chr4**	**-**	170,126,345	170,133,883
**chimp**	**chr6**	**+**	48,375,302	48,381,314
**human**	**chr6**	**+**	48,873,675	48,879,725
**chimp**	**chr7**	**-**	147,138,951	147,141,648
**human**	**chr7**	**-**	150,561,277	150,563,994
**chimp**	**chr8**	**-**	43,516,710	43,525,262
**human**	**chr8**	**-**	43,694,016	43,702,583
**chimp**	**chr8**	**+**	76,757,424	76,759,791
**human**	**chr8**	**+**	78,652,302	78,654,820
**chimp**	**chr13**	**+**	66,107,993	66,115,813
**human**	**chr13**	**+**	84,869,526	84,877,320
**chimp**	**chr15**	**-**	23,660,410	23,668,998
**human**	**chr15**	**-**	45,234,477	45,243,073
**chimp**	**chr16**	**-**	19,355,716	19,364,289
**human**	**chr16**	**-**	19,393,581	19,402,152
**chimp**	**chr19**	**+**	39,063,080	39,068,105
**human**	**chr19**	**+**	40,954,172	40,959,178
chimp ^#^	chr5	-	60,565,330	60,572,694
chimp	chrY	+	1,786,750	1,792,226
chimp	chrY	-	2,019,892	2,025,806
chimp	chrY	-	4,753,348	4,754,918
chimp	chrY	+	5,360,268	5,361,838
chimp	chrY	+	6,151,270	61,571,65
chimp	chrY	+	8,716,313	8,718,506
chimp	chrY	-	9,765,071	9,765,511
chimp	chrY	-	10,496,235	10,498,083
chimp	chrY	+	10,590,060	10,592,276
chimp	chrY	-	10,822,682	10,824,473
chimp	chrY	-	13,423,428	13,425,003
chimp	chrY	+	13,999,593	14,001,168
chimp	chrY	+	14,791,990	14,797,426
chimp	chrY	+	16,516,529	16,520,265
chimp	chrY	-	16,748,108	16,753,037
human	chr1	-	155,629,408	155,632,775
human	chr5	-	156,660,448	156,663,815
human	chr8	+	145,019,974	145,032,719
human	chr10	-	99,822,511	99,825,532
human	chr12	+	48,509,228	48,511,681
human	chr21	-	18,563,368	18,566,735
human	chrY	-	8,121,821	8,126,768
human	chrY	+	8,996,062	9,000,755
human	chrY	-	9,273,707	9,279,611
human	chrY	-	17,669,948	17,671,523
human	chrY	-	18,622,534	18,626,952
human	chrY	+	21,580,120	21,585,551
human	chrY	-	21,845,475	21,850,069

***** Bold font represents homologous elements between human and chimpanzee. ^#^ Non-bold font represents non-homologous elements. The background color is used to distinguish homologous pairs.

**Table 4 viruses-16-00892-t004:** The comparison of chimpanzee HML-9 solo LTRs and their counterparts in the human genome.

Species	Chromosome	Strand	Position Start	Position End
**chimp ***	**chr1**	**+**	143,492,646	143,493,186
**human ***	**chr1**	**+**	168,457,190	168,457,732
**chimp**	**chr2A**	**+**	78,001,602	78,002,185
**human**	**chr2**	**+**	77,807,602	77,808,185
**chimp**	**chr2A**	−	81,498,383	81,498,970
**human**	**chr2**	−	81,304,430	81,305,068
**chimp**	**chr3**	**+**	44,753,258	44,753,829
**human**	**chr3**	**+**	44,534,488	44,535,059
**chimp**	**chr3**	−	59,703,426	59,704,196
**human**	**chr3**	−	59,469,489	59,470,030
**chimp**	**chr3**	−	81,878,200	81,878,786
**human**	**chr3**	−	81,329,902	81,330,488
**chimp**	**chr3**	−	84,155,763	84,156,349
**human**	**chr3**	−	83,618,409	83,618,995
**chimp**	**chr3**	−	161,301,797	161,302,385
**human**	**chr3**	−	163,283,189	163,283,777
**chimp**	**chr4**	**+**	19,038,454	19,039,041
**human**	**chr4**	**+**	19,556,097	19,556,684
**chimp**	**chr5**	−	38,971,726	38,972,307
**human**	**chr5**	**+**	75,859,521	75,860,102
**chimp**	**chr5**	−	84,066,356	84,066,944
**human**	**chr5**	−	105,998,962	105,999,549
**chimp**	**chr6**	−	79,799,818	79,800,406
**human**	**chr6**	−	82,297,755	82,298,498
**chimp**	**chr6**	**+**	166,662,836	166,663,420
**human**	**chr6**	**+**	169,084,226	169,084,808
**chimp**	**chr7**	**+**	14,600,052	14,600,637
**human**	**chr7**	**+**	14,509,240	14,509,827
**chimp**	**chr7**	**+**	140,435,975	140,436,547
**human**	**chr7**	**+**	143,472,173	143,472,744
**chimp**	**chr8**	−	49,114,425	49,115,012
**human**	**chr8**	−	51,178,592	51,179,179
**chimp**	**chr10**	−	19,329,882	19,330,469
**human**	**chr10**	−	18,856,645	18,857,233
**chimp**	**chr11**	**+**	120,039,148	120,039,717
**human**	**chr11**	**+**	124,270,705	124,271,275
**chimp**	**chr12**	−	50,879,963	50,880,545
**human**	**chr12**	**+**	38,144,469	38,145,052
**chimp**	**chr13**	−	19,353,563	19,354,142
**human**	**chr13**	−	38,319,721	38,320,300
**chimp**	**chr14**	**+**	19,027,337	19,028,300
**human**	**chr14**	**+**	38,011,040	38,012,012
**chimp**	**chr14**	**+**	28,857,266	28,857,843
**human**	**chr14**	**+**	48,011,215	48,011,780
**chimp**	**chr15**	**+**	17,411,474	17,412,062
**human**	**chr15**	**+**	39,011,033	39,011,621
**chimp**	**chr18**	**+**	10,395,647	10,396,231
**human**	**chr18**	−	4,527,618	4,528,209
**chimp**	**chr21**	−	979,256	979,845
**human**	**chr21**	−	14,080,466	14,081,052
**chimp**	**chrX**	**+**	10,991,573	10,992,157
**human**	**chrX**	**+**	11,033,746	11,034,330
**chimp**	**chrX**	−	88,705,817	88,706,830
**human**	**chrX**	−	93,273,183	93,274,197
**chimp**	**chrX**	−	138,003,751	138,004,333
**human**	**chrX**	−	142,767,872	142,768,454
chimp ^#^	chr1	+	137,442,976	137,443,751
chimp	chr2B	−	11,387,988	11,388,572
chimp	chr2B	+	22,085,802	22,086,389
chimp	chr2B	−	25,439,654	25,440,248
chimp	chr2B	+	27,967,992	27,968,572
chimp	chr2B	+	66,792,632	66,793,222
chimp	chr2B	−	80,853,578	80,854,165
chimp	chr7	−	48,429,668	48,430,256
chimp	chr16	−	31,333,554	31,334,129
chimp	chrY	−	18,630,426	18,631,014
human	chr1	+	162,419,359	162,419,942
human	chr2	+	97,964,920	97,965,508
human	chr2	−	125,024,208	125,024,792
human	chr2	+	135,521,883	135,522,470
human	chr2	−	138,860,917	138,861,512
human	chr2	+	171,365,032	171,365,617
human	chr2	+	180,236,847	180,237,437
human	chr2	−	194,256,159	194,256,746
human	chr3	−	154,944,330	154,944,911
human	chr4	+	116,980,222	116,980,809
human	chr4	+	129,080,872	129,081,454
human	chr11	−	62,185,237	62,185,824
human	chr12	−	4,720,007	4,720,593
human	chr15	−	54,594,796	54,595,373
human	chr17	+	52,961,655	52,962,071
human	chr18	+	63,648,105	63,648,555
human	chr20	−	2,809,052	2,809,886
human	chrX	−	144,791,258	144,791,846
human	chrY	−	25,974,734	25,975,320

***** Bold font represents homologous elements between human and chimpanzee. ^#^ Non-bold font represents non-homologous elements.

**Table 5 viruses-16-00892-t005:** The integrity of six separate regions relative to the corresponding sections of reference.

Number	Provirus Regions	5′LTR (%)	Gag (%)	Pro (%)	Pol (%)	Env (%)	3′LTR (%)
1	chr16 19,355,716 19,364,289	100.00%	99.61%	99.89%	99.43%	99.17%	99.66%
2	chr2A 82,212,507 82,221,098	98.98%	99.66%	100.00%	99.47%	99.90%	99.15%
3	chr8 43,516,710 43,525,262	99.66%	99.83%	98.20%	99.40%	98.83%	99.83%
4	chr15 23,660,410 23,668,998	99.83%	99.22%	99.77%	99.51%	99.46%	99.83%
5	chr13 66,107,993 66,115,813	35.95%	98.49%	52.25%	99.66%	99.76%	99.66%
6	chr5 60,565,330 60,572,694	73.42%	99.72%	99.55%	75.25%	48.48%	98.64%
7	chr6 48,375,302 48,381,314	98.47%	92.91%	61.71%	6.04%	99.85%	98.98%
8	chrY 2,019,892 2,025,806	88.25%	91.74%	64.86%	6.19%	99.56%	99.15%
9	chrY 6,151,270 6,157,165	83.82%	91.68%	65.32%	6.04%	99.56%	99.83%
10	chr19 39,063,080 39,068,105	98.98%	99.22%	64.30%	6.11%	67.35%	79.56%
11	chrY 1,786,750 1,792,226	88.59%	91.74%	64.86%	6.19%	99.56%	22.83%
12	chrY 14,791,990 14,797,426	88.59%	91.74%	64.86%	6.19%	99.56%	16.52%
13	chrY 16,748,108 16,753,037	0.00%	91.35%	64.86%	6.19%	99.56%	99.83%
14	chr4 168,066,681 168,071,528	0.00%	92.69%	99.77%	87.66%	0.00%	0.00%
15	chrY 16,516,529 16,520,265	0.00%	64.99%	64.86%	6.19%	93.70%	0.00%
16	chr7 147,138,951 147,141,648	0.00%	0.00%	0.00%	16.64%	99.12%	51.45%
17	chr8 76,757,424 76,759,791	0.00%	0.00%	0.00%	0.00%	86.95%	100.00%
18	chrY 10,590,060 10,592,276	56.90%	91.74%	7.77%	0.00%	0.00%	0.00%
19	chrY 8,716,313 8,718,506	0.00%	0.00%	0.00%	6.11%	99.32%	10.22%
20	chrY 10,822,682 10,824,473	0.00%	0.00%	0.00%	0.00%	58.26%	99.66%
21	chrY 13,999,593 14,001,168	52.81%	62.09%	0.00%	0.00%	0.00%	0.00%
22	chrY 5,360,268 5,361,838	52.47%	61.92%	0.00%	0.00%	0.00%	0.00%
23	chrY 13,423,428 13,425,003	52.64%	62.14%	0.00%	0.00%	0.00%	0.00%
24	chrY 4,753,348 4,754,918	51.96%	62.09%	0.00%	0.00%	0.00%	0.00%
25	chrY 10,496,235 10,498,083	0.00%	0.00%	0.00%	0.00%	86.41%	9.20%
26	chrY 9,765,071 9,765,511	0.00%	0.00%	0.00%	0.00%	21.31%	0.00%

**Table 6 viruses-16-00892-t006:** Estimated time of HML-9 elements integration.

Species	Provirus Regions	Divergence from Consensus Sequence				Mean Divergences	T = D/0.2	Age/Million Years (Gene vs. Consensus)	Divergence between 2 LTRs	T = D/0.2/2	Age/Million Years (LTR vs. LTR)
		Gag	Pro	Pol	Env						
**chimp ***	**chr2A 82,212,507 82,221,098**	**0.057**	**0.060**	**0.063**	**0.085**	**0.066**	**0.33**	**33.13**	**0.057**	**0.14**	**14.25**
**human ***	**chr2 82,022,660 82,031,279**	**0.061**	**0.182**	**0.204**	**0.101**	**0.137**	**0.69**	**68.50**	**0.070**	**0.18**	**17.50**
**chimp**	**chr4 168,066,681 168,071,528**	**0.051**	**0.063**	**NA**	**NA**	**0.057**	**0.29**	**28.50**	**NA**	**NA**	**NA**
**human**	**chr4 170,126,345 170,133,883**	**0.058**	**0.172**	**0.214**	**NA**	**0.148**	**0.74**	**74.00**	**NA**	**NA**	**NA**
**chimp**	**chr6 48,375,302 48,381,314**	**0.091**	**NA**	**NA**	**0.078**	**0.085**	**0.42**	**42.25**	**0.106**	**0.27**	**26.00**
**human**	**chr6 48,873,675 48,879,725**	**0.063**	**NA**	**NA**	**0.106**	**0.085**	**0.42**	**42.25**	**0.110**	**0.28**	**27.50**
**chimp**	**chr7 147,138,951 147,141,648**	**NA**	**NA**	**NA**	**0.235**	**0.235**	**1.18**	**117.50**	**NA**	**NA**	**NA**
**human**	**chr7 150,561,277 150,563,994**	**NA**	**NA**	**NA**	**0.197**	**0.197**	**0.99**	**98.50**	**NA**	**NA**	**NA**
**chimp**	**chr8 43,516,710 43,525,262**	**0.072**	**0.076**	**0.071**	**0.086**	**0.076**	**0.38**	**38.13**	**0.103**	**0.26**	**25.75**
**human**	**chr8 43,694,016 43,702,583**	**0.091**	**0.177**	**0.231**	**0.121**	**0.155**	**0.78**	**77.50**	**0.107**	**0.27**	**26.75**
**chimp**	**chr13 66,107,993 66,115,813**	**0.058**	**NA**	**0.055**	**0.070**	**0.061**	**0.31**	**30.50**	**NA**	**NA**	**NA**
**human**	**chr13 84,869,526 84,877,320**	**0.054**	**NA**	**0.208**	**0.103**	**0.122**	**0.61**	**60.83**	**NA**	**NA**	**NA**
**chimp**	**chr15 23,660,410 23,668,998**	**0.056**	**0.054**	**0.046**	**0.063**	**0.055**	**0.27**	**27.38**	**0.076**	**0.19**	**19.00**
**human**	**chr15 45,234,477 45,243,073**	**0.051**	**0.126**	**0.206**	**0.099**	**0.121**	**0.60**	**60.25**	**0.080**	**0.20**	**20.00**
**chimp**	**chr16 19,355,716 19,364,289**	**0.053**	**0.038**	**0.053**	**0.063**	**0.052**	**0.26**	**25.88**	**0.090**	**0.23**	**22.00**
**human**	**chr16 19,393,581 19,402,152**	**0.059**	**0.158**	**0.206**	**0.089**	**0.128**	**0.64**	**64.00**	**0.082**	**0.21**	**20.50**
**chimp**	**chr19 39,063,080 39,068,105**	**0.045**	**NA**	**NA**	**NA**	**0.045**	**0.23**	**22.50**	**0.109**	**0.27**	**27.25**
**human**	**chr19 40,954,172 40,959,178**	**0.075**	**NA**	**NA**	**NA**	**0.075**	**0.38**	**37.50**	**0.097**	**0.24**	**24.25**
chimp ^#^	chr5 60,565,330 60,572,694	0.061	0.056	NA	NA	0.059	0.29	29.25	0.089	0.22	22.25
chimp	chrY 1,786,750 1,792,226	0.115	NA	NA	0.099	0.107	0.54	53.50	NA	NA	NA
chimp	chrY 2,019,892 2,025,806	0.113	NA	NA	0.100	0.107	0.53	53.25	0.144	0.36	36.00
chimp	chrY 6,151,270 6,157,165	0.114	NA	NA	0.098	0.106	0.53	53.00	0.144	0.36	36.00
chimp	chrY 8,716,313 8,718,506	NA	NA	NA	0.100	0.100	0.50	50.00	NA	NA	NA
chimp	chrY 10,590,060 10,592,276	0.114	NA	NA	NA	0.114	0.57	57.00	NA	NA	NA
chimp	chrY 14,791,990 14,797,426	0.114	NA	NA	0.099	0.107	0.53	53.25	NA	NA	NA
chimp	chrY 16,516,529 16,520,265	NA	NA	NA	0.097	0.097	0.49	48.50	NA	NA	NA
chimp	chrY 16,748,108 16,753,037	0.112	NA	NA	0.098	0.105	0.53	52.50	NA	NA	NA
human	chr1 155,629,408 155,632,775	NA	0.212	0.156	NA	0.184	0.92	92.00	NA	NA	NA
human	chr5 156,660,448 156,663,815	NA	0.215	0.160	NA	0.188	0.94	93.75	NA	NA	NA
human	chr8 145,019,974 145,032,719	NA	NA	0.513	0.093	0.303	1.52	151.50	NA	NA	NA
human	chr10 99,822,511 99,825,532	NA	NA	0.169	NA	0.169	0.85	84.50	NA	NA	NA
human	chr21 18,563,368 18,566,735	NA	0.266	0.190	NA	0.228	1.14	114.00	NA	NA	NA
human	chrY 8,121,821 8,126,768	NA	NA	NA	0.125	0.125	0.63	62.50	0.157	0.39	39.25
human	chrY 8,996,062 9,000,755	NA	NA	NA	0.133	0.133	0.67	66.50	NA	NA	NA
human	chrY 9,273,707 9,279,611	0.114	NA	NA	0.139	0.127	0.63	63.25	0.141	0.35	35.25
human	chrY 18,622,534 18,626,952	NA	NA	NA	NA	NA	NA	NA	0.194	0.49	48.50
human	chrY 21,580,120 21,585,551	0.105	NA	NA	0.140	0.123	0.61	61.25	NA	NA	NA
human	chrY 21,845,475 21,850,069	NA	NA	NA	0.143	0.143	0.72	71.50	NA	NA	NA

***** Bold font represents homologous elements between human and chimpanzee. ^#^ Non-bold font represents non-homologous elements. The background color is used to distinguish homologous pairs.

## Data Availability

Data are contained within the article and Appendix A; further inquiries can be directed to the corresponding authors.

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
