# Peer review of "Comprehensive Identification and Characterization of HML-9 Group in Chimpanzee Genome"

_viruses, 2024, doi:10.3390/v16060892_

Round 1

Reviewer 1 Report

Comments and Suggestions for Authors

Comments on the Quality of English Language

See my overall comments.  The quality of English is unacceptable.

Author Response

The content of the response is contained in the document.

Reviewer 2 Report

Comments and Suggestions for Authors

Summary

In this work, Yang et al performed a comprehensive analysis of ERV K (HML-9) group of endogenous retroviruses in the draft chimpanzee genome. The strategies and approaches used to characterize the insertions are standard for the field and are applied appropriately. Overall, this is solid work that adds to our knowledge of endogenous retrovirus evolution in primates.

Specific comments

1. Are any of the insertions polymorphic in the chimp genome?

2. Is anything known about HML-9 expression either in chimps or humans, especially for insertions in the Y chromosome?

3. Is anything known about the target site duplications resulting from the original insertion of “complete” proviruses or solo LTRs? Note that solo LTRs should still have TSDs if they were formed by intra-element homologous recombination. Also, polymorphic HML-2 insertions have unusually long TSDs. Perhaps this property is conserved.

Comments on the Quality of English Language

English needs improvement.

Author Response

(The authors gave the same response as above.)

Reviewer 3 Report

Comments and Suggestions for Authors

This manuscripts catalogs endogenous retroviruses (ERVs) in the publicly available chimpanzee genome. They focus on the youngest ERV family equivalent to HERV-K in humans. Homology-based search is used, an results in detection of 26 proviruses and 38 remnants (solitary LTRs). The genomic structure of these elemennts was determined, chromosomal distribution, phylogeny, evolutionary age,  and closeness to neighboring genes. Also, motifs enriched in LTR and primer binding site motifs are described. Overall, this is a quite comprehensive study.

Comments:

1.       My main objection is insufficient description and verification of the phylogenetic age of the individual ERV elements. The authors use well established comparison of viral LTR ends. They also use internal sequences (gag, pol , env) in a way that is not well described and yields very different timeframes. Importantly, a key analysis is missing, describing corresponding HERV-K integration sites in humans. For such a small number of integrations, they can all be analyzed. The specific chromosomal locus can be either “empty”, hinting at integration that followed chimp-human speciation, or those integration should be shared, meaning integration before the speciation event. This is a key analysis that should be added.

2.       Fig 1 A – there seem to be too many integration shown compared to what is described in text.

3.       The large tables should be put into supplementary data.

4.       In many figures, the font is too small to read, please reformat.

Comments on the Quality of English Language

editing is needed

Author Response

(The authors gave the same response as above.)

Reviewer 4 Report

Comments and Suggestions for Authors

The manuscript by Yang et al. describes the identification, location and integration time of the HERV-K group HML9. This group performed a similar publication with human HML9 (see ref. 47).

Several points need to be clarified or added:

1) Under 3.1 the authors claim that HML9 was not detected on chromosome 1, 2B, 3, 9 etc. In Discussion (p.7) they claim “…ERV-K HML-9 is distributed on every chromosome… What is correct?

2) The authors analyzed the proviral ERV for full length sizes. Important would be, which HML9 would be codogenic for gag, pol, env; esp. full-length genes. Please use ORF analyses.

3) The authors also analyzed the human HML9 integrations. It would be helpful to conclude the CERV and HERV in one table for chromosome location and length. Important would be chromosome 9 and 22, which are devoid of HML9 in humans.

4) The CERV LTR (LTR14C) integrated 25.5 mya, the HERV 28.83 mya. Considering the human-chimpanzee split for about 7 mya, please explain the discrepancy. On the other hand, please show with the help of the new table (see 3) differences of integration (or rather new translocations) and mutations between human and chimpanzees, which should have happened esp. in the last 7 mya.

5) Fig. 3, 6 and 8 are too small and actually only decipherable after zooming in. Please split the figures for better readability. If length is a problem, you can reduce 3.6, including Fig. 5, 6, 7 and 8, which is only very speculative (without any proofs) and does not help the topic of ERV integration. More important would be a point to point contrasting CERV and HERV.

Comments on the Quality of English Language

Minor revisions (e.g. 3.5. "For DNA, moti refers to the DNA binding site...")

Author Response

(The authors gave the same response as above.)

Round 2

Reviewer 1 Report

Comments and Suggestions for Authors

Overall, there are improvements, but some of the major and minor problems from the original manuscript remain.

Major Issues

I am trying very hard to ignore the combined lack of understanding and condescension displayed by many of the authors’ responses. Repeated copying and pasting of descriptions of procedures like BLAT from some textbook source is not responsive to the reviewers’ attempts to help the authors clean up serious problems present in the manuscript as submitted, and does not substitute for correcting the flaws in the manuscript itself.  Although the English has largely been corrected to an acceptable standard, and some figures and tables have had helpful addition of the human homologs, many errors in logic and interpretation persist, as do inclusion of meaningless analyses, and I still cannot recommend this resubmission for publication.

First, the nonsensical idea that the same set of proviruses has different ages in humans and chimpanzees  (28.83 mya vs 45.33 mya) must be corrected by confining the analysis to orthologous 2-LTR proviruses at the same sites in the 2 species only.  If the results are still discrepant, it can only mean that the calculation is incorrect for some reason. Proviruses that have been reduced to solo LTRs in one or the other species must NOT be included, but rather discussed separately.  For Table 4, the authors need to include all human homologs in the calculations then show which specific proviruses are causing the difference in calculated integration time that they're observing. They need to mark out the individual ones that are solo-LTRs instead of 2-LTR integrations differentially between humans and chimpanzees, or that are differentially rearranged post integration, duplicated, etc.

Further, the calculations of ages require a statistical analysis to show whether any observed or calculated differences are significant.

Please provide ages confidence intervals for the p values for the individual age differences as calculated.

Second, although they now include comparisons with their human homologs, the authors still fail to use these to draw some interesting conclusions regarding evolution of the relationship between them: Points they should discuss include:

1.         The relative rates of solo LTR formation over the time since the last common ancestor, as compared to the time since integration.

2.        The evident multiple duplication of a segment of the chimpanzee Y chromosome contaning the same provirus (Table 3, Lines 21-24), from their divergence, it should be possible to estimate how long ago and in what order these events happened.

3.        The rate of solo lTR formation on the Y chromosome relative to autosomes.  If much lower, that would imply that solo LTR formation arose during meiotic recombination between chromosome pairs.

Third, the most important figures 2 and 3 are so tiny that they are completely useless.  Figure 3, in particular, should occupy at least one full page, perhaps more. 

Fourth, the MEME analysis (section 3.5) conveys no information of value to the reader and should be deleted, or, at most, moved to a supplement.  The e values, sone smaller than 10-200 are nonsensical.  They are vastly smaller than the probability of finding exact matches in unrelated sequences of length 35 (about 10-21). 

Figure 5 and 6 report exactly the same analysis as published in reference 47 and should have exactly the same result, and therefore not be published.

Other Points:

In the authors’ response:

1. the 3x copy-pasted description of BLAT, which references studies that used BLAT in primates for completely different analyses, is completely off the point. Nobody's arguing that BLAT can't be a useful tool in primates, the question is whether it is appropriate for this specific analysis. The block of text about BLAT doesn't answer that question. The limitations of BLAT for this analysis are not commented on by the authors in the manuscript. The only other use of BLAT for identification of HERVs specifically, referenced by these authors in this response, are from another of their own papers.

2. From the copy-pasted block:

“Based on our experience, this tool is sufficient for our work and misidentification of proviruses is not one of the possible reasons. Thank you very much. “ does not address the concern regarding using BLAT for HERV identification where RepeatMasker/RepeatModeler is more standard and typical and does not actually test whether misidentification has or has not occurred. It’s also very patronizing.

Regarding the similarity of ERVs and retroviruses:

3.                          Response: “We don't approve of your idea. The genetic composition of retroviruses from 5’-3’ includes LTR-gag-pol-env-LTR, and the structural composition of each retrovirus is different, such as HIV, HTLV, and ERV. The gene structure of HIV consists of LTR,3 structural genes (gag, pol and env) and 6 regulatory genes (vpr, rev, vif, tat, vpu/vpx, nef). The gene structure of ERV is LTR, gag, pol, env and LTR. The gene structure of HTLV consists of two LTR, 3 structural genes (gag, pol and env) and two regulatory genes (tax and rex). Based on the above, we use "similar" instead of "identical" in the manuscript.” Perhaps this response is due to a misunderstanding due to a language barrier, but the original comment was not that “all retroviruses are identical” which is obviously false, but that HERVs are in fact identical to retroviruses, because they originally were exogenous retroviruses. It’s also a very odd, and incorrect, statement about ERVs. “ERV” is not a single retrovirus, and although a given ERV’s ancestral exogenous retrovirus must have contained LTRs, gag, pol, env, it also must have contained pro (part of the gag-pro-pol polyprotein), like all retroviruses, and different ERVs can and do encode accessory proteins (e.g., rec), like any other retrovirus. They also forgot pro in HTLV and HIV, in their response.

4.                          Our comment “incomplete elements containing many gaps were excluded manually”  The process and criteria for exclusion need to be specified here.

was not addressed at all, instead, the copy-pasted block about BLAT was included again.

5.        Our comment: Please limit the age estimates to 2 significant figures.  4 is absurd.

Response: Thank you very much for your valuable comments. Age estimate retains 2 decimal places.

Do the authors not know the difference between “significant figures” and “decimal places”?  There should be no decimal places in any of the ages. And there should be statistics to show the precision and significance of the calculated numbers.

In the manuscript:

Line 16-17 ERVs are not “retrotransposons”

Line 52-53 What is the “first amino acid bound … by the PBS?” Makes no sense.

Line 268 Distribution, not “integration”

Line 276 how can a provirus be “in introns and exons?”

Line 339 “group, and”. Something is missing

Comments on the Quality of English Language

There are still a few errors. Please go over carefully and correct.

Reviewer 2 Report

Comments and Suggestions for Authors

The authors have successfully addressed my questions and I suggest accepting the revised manuscript for publication

Comments on the Quality of English Language

Requires moderate editing.

Reviewer 3 Report

Comments and Suggestions for Authors

The authors added the comparison of chimpanzee ERVs and their counterparts in the human genome in table 1. I am missing one last thing: there should be clear discussion about the number of ERVs that were/were not detected in both genomes. This should include both the full and single LTR forms. The evolutionary age span determined for each chimpanzee ERVs should be compared to the estimated age of human/chimpanzee split. I do not see this clearly stated and discussed in the current manuscript.

Comments on the Quality of English Language

minor edits needed

Reviewer 4 Report

Comments and Suggestions for Authors

After downloading the "revised" manuscript no tables or figures were integrated in the text. Are they the same like in version 1? If yes, the figures have to be refined. I still think that paragraph 3.6 can be shortened and improved with a table contrasting CERV and HERV.

Comments on the Quality of English Language

no comments
